# INTERACTION-CONSISTENT OBJECT REMOVAL VIA MLLM-BASED REASONING

(i) Lighting-dependent effects
"Remove the person."

(ii) Physically connected objects
"Remove the dog."

(iii) Target-produced elements
"Remove the person."

(iv) Contextually linked objects
"Remove the charging station."

Input Image    Ours    Input Image    Ours

Figure 1: We introduce the task of Interaction-Consistent Object Removal (ICOR), which aims to remove not only the target object itself but also its associated interaction elements, in order to produce semantically coherent and visually realistic outputs. These interaction effects can be broadly categorized into four types: lighting-dependent effects, physically connected objects, target-produced elements, and contextually linked objects.

## ABSTRACT

Image-based object removal often erases only the named target, leaving behind interaction evidence that renders the result semantically inconsistent. We formalize this problem as Interaction-Consistent Object Removal (ICOR), which requires removing not only the target object but also associated interaction elements, such as lighting-dependent effects, physically connected objects, target-produced elements, and contextually linked objects. To address this task, we propose Reasoning-Enhanced Object Removal with MLLM (REORM), a reasoning-enhanced object removal framework that leverages multimodal large language models to infer which elements must be jointly removed. REORM features a modular design that integrates MLLM-driven analysis, mask-guided removal, and a self-correction mechanism, along with a local-deployment variant that supports accurate editing under limited resources. To support evaluation, we introduce ICOREval, a benchmark consisting of instruction-driven removals with rich interaction dependencies. On ICOREval, REORM outperforms state-of-the-art image editing systems, demonstrating its effectiveness in producing interaction-consistent results.

# 1 INTRODUCTION

Object removal is a fundamental task in image editing, supporting a wide range of real-world applications. Its goal is to remove a target object from an image and inpaint the missing region to produce a visually realistic result. While recent advances in object removal (Jiang et al. (2025); Ekin et al. (2024); Sun et al. (2025)) and instruction-guided image editing (Fu et al. (2024); Huang et al. (2024)) have leveraged powerful diffusion models and multimodal large language models (MLLMs) to improve controllability and visual quality, they largely focus on removing the target object itself without reasoning about its interaction with the surrounding scene. Some recent methods have begun to address specific visual effects such as shadows and reflections, but their scope remains limited and does not cover broader forms of object–scene interactions (Winter et al. (2024); Wei et al. (2025); Zhao et al. (2025)). As a result, associated elements or contextually dependent objects are often ignored, leading to semantic inconsistencies in the edited outputs.

To address this limitation, we introduce the notion of *Interaction-Consistent Object Removal* (ICOR), which extends traditional object removal by explicitly considering object-scene interactions. ICOR demands a holistic understanding of the entire image as well as the interaction relationships among objects. When instructed to remove a specific object, the process goes beyond simply eliminating the target itself and instead infers the plausible appearance of the scene in its absence. This involves identifying and removing all secondary elements that would become inconsistent or illogical once the target object is removed, with the goal of producing coherent and realistic results.

We categorize interaction effects into four types: lighting-dependent effects, physically connected objects, target-produced elements, and contextually linked objects. Fig. 1 illustrates an example for each type. In (i), when removing the person, their shadow must also be removed. In (ii), removing the dog requires also removing the leash it is wearing. **In (iii), if the person is removed, the tonal shifts in the wet sand caused by their weight and the remaining footprints become logically inconsistent.** In (iv), removing the charging station renders the adjacent signage semantically irrelevant, as it refers to an object that no longer exists. To preserve scene coherence, the sign should be removed as well.

In this paper, we propose Reasoning-Enhanced Object Removal with MLLM (REORM), a framework designed to address the ICOR task. The key idea behind REORM is to leverage the broad commonsense reasoning ability of MLLM to analyze the implications of removing a target object. Given an instruction to remove an object, the MLLM is employed to infer which additional elements in the scene would become semantically inconsistent or physically incorrect if the target were absent. These associated elements are then identified and grouped as removal candidates.

To implement this idea, we design REORM as a modular framework. In the main stage, MLLM-driven analysis and removal, an MLLM interprets the instruction, identifies the target object, and performs reasoning to infer associated elements that must also be removed for consistency. These elements are then located using open-vocabulary segmentation and removed with a mask-guided object removal model. To support local deployment, we adopt a prompt chaining strategy and distribute the reasoning process across a compact MLLM and LLM. Finally, to improve output quality, we introduce an MLLM-controlled self-correction stage that simulates the expected result, compares it with the edited image, and resolves remaining inconsistencies. REORM combines strong commonsense reasoning with modular flexibility and practical deployability, enabling interaction-consistent object removal across diverse scenes without additional training.

Since ICOR is a newly introduced task, we further construct a new benchmark, ICOREval, consisting of paired images, instructions, and ground-truth results. Experimental evaluations on this benchmark demonstrate that REORM achieves superior performance in handling interaction-consistent cases compared to existing methods. In summary, our key contributions are as follows:

• We introduce Interaction-Consistent Object Removal (ICOR), a new task that extends conventional object removal by explicitly addressing object interactions and contextual dependencies.

• We propose Reasoning-Enhanced Object Removal with MLLM (REORM), a framework that leverages LLMs for commonsense reasoning to address the ICOR task.

• We present ICOREval, the first benchmark for ICOR. Experimental results show that REORM achieves superior performance in terms of image quality and robustness.

## 2 RELATED WORK

**Instruction-based Image Editing** aims to modify an image according to natural language commands. Early instruction-based editing approaches, ranging from GAN-based to diffusion-based methods, have been extensively surveyed in Nguyen et al. (2024). More recent methods leverage multimodal large language models (MLLMs) to enhance semantic understanding and controllability. Fu et al. (2024) use an MLLM to rewrite and ground user instructions before guiding the editing process, while Huang et al. (2024) combine an MLLM with a bidirectional interaction module to reason over complex instructions and localize edits more effectively. Despite incorporating MLLMs, these methods do not fully utilize the MLLM's commonsense reasoning capabilities to model interaction relationships. As a result, they fail to modify associated elements, limiting their ability to produce interaction-consistent edits.

**Object Removal** has been widely studied as an image editing task, with most methods relying on inpainting within user-provided masks. Diffusion-based editors have advanced large-mask completion (Banitalebi-Dehkordi & Zhang (2021), Rombach et al. (2022)), while recent tuning-free or plug-in removers leverage internal visual features or attention mechanisms to guide object removal (Ekin et al. (2024), Sun et al. (2025), Jiang et al. (2025)). Most of these methods operate primarily within the boundaries of the provided mask.

Recent works have begun to incorporate object-effect removal, aiming to eliminate not only the target object but also its lighting-dependent effects, such as shadows and reflections. Winter et al. (2024) takes a supervised, counterfactual approach by capturing the same scene before and after removing a single object and fine-tuning a diffusion model. Wei et al. (2025) enhances object–background separation with explicit guidance to suppress residual artifacts. Zhao et al. (2025) introduces object-effect attention to jointly erase objects and their induced effects. However, their scope remains limited to lighting-dependent effects, and they lack explicit modeling of higher-level interaction relationships such as physically connected and contextually linked objects. This limitation motivates our formulation of interaction-consistent object removal, which addresses a broader range of interactions beyond mask and effect boundaries.

**Layer Decomposition** provides an alternative route to object removal: by explicitly separating a scene into editable layers, the target object and its associated effects can be isolated and removed while preserving the background. Lu et al. (2021) estimate RGBA layers that bundle a subject together with its time-varying effects (e.g., shadows, reflections, smoke), enabling object removal or recompositing in videos that preserves these effects. Lee et al. (2025) extends this idea with stronger generative priors and multi-layer outputs for more controllable video editing. These video-based decomposition approaches rely on temporal cues. For still images, Yang et al. (2024) employs a diffusion-based model trained with simulated layered data and recomposition loss to separate a clean background and a transparent foreground that preserves shadows and reflections. However, while effective for preserving lighting-dependent effects and smoke, these approaches do not model higher-level interaction relationships, such as physical connections or contextual dependencies.

## 3 TASK OVERVIEW: INTERACTION-CONSISTENT OBJECT REMOVAL

We introduce Interaction-Consistent Object Removal (ICOR), which extends conventional object removal by requiring models to account for interactions and dependencies among scene elements. For example, removing a dog being walked should also remove the leash, shadow, and footprints; otherwise, the result is inconsistent. ICOR therefore challenges vision systems to holistically model object–scene relationships and produce edits that are both coherent and realistic.

**Task Definition.** Given an input image $I \in \mathbb{R}^{H \times W \times 3}$ and an instruction $T$, the goal of ICOR is to produce an edited image $\hat{I}$, where the target object, along with all elements rendered inconsistent by its removal, are eliminated while maintaining overall scene plausibility, $\text{ICOR} : (I, T) \mapsto \hat{I}$.

We define *target objects* as the entities explicitly mentioned in the instruction, and *associated elements* as all elements that interact with the target objects.

**Scope.** The associated elements considered in ICOR fall into four categories:

(i) Lighting-dependent effects – visual artifacts arising from the interaction of the target object with scene lighting, such as shadows or reflections.

(ii) Physically connected objects – items held, used, or otherwise physically supported by the target, such as a bag carried in hand.

(iii) Target-produced elements – tangible traces directly created by the target, such as footprints on sand.

(iv) Contextually linked objects – elements that lose their purpose or context without the target; for example, a fire extinguisher sign with no extinguisher.

Given that such interactions can be nested or chained, recursive reasoning is necessary to identify all elements that should be removed to ensure a coherent result.

## 4  METHOD: REASONING-ENHANCED OBJECT REMOVAL WITH MLLM

We propose the Reasoning-Enhanced Object Removal with MLLM (REORM) framework, as illustrated in Fig. 2. The core of our framework lies in MLLM-driven analysis and removal (Section 4.1), where an MLLM performs reasoning to identify the target object along with its associated elements, and then guides the removal process. We further introduce MLLM-controlled self-correction (Section 4.2), a mechanism to refine the edited image through an additional round of reasoning and verification. Finally, we implement MLLM–LLM Collaboration for Local Deployment (Section 4.3), a strategy designed to address the reduced reasoning capabilities of smaller MLLMs.

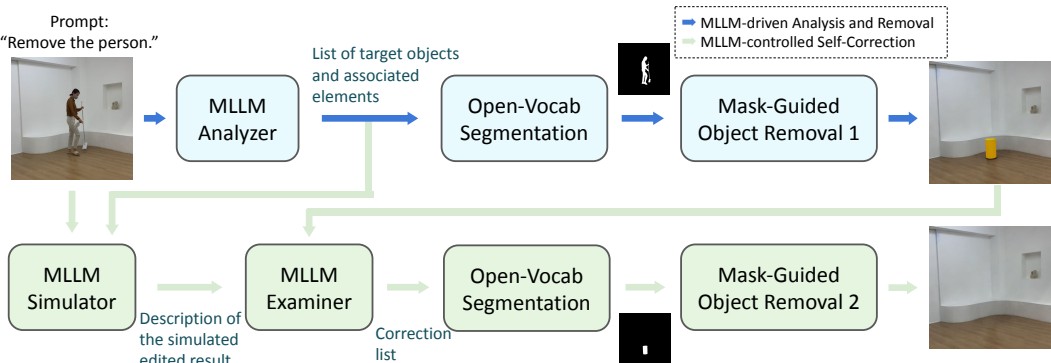

Figure 2: Overview of the proposed REORM, which leverages the commonsense reasoning capabilities of an MLLM to enable interaction-consistent object removal.

### 4.1  MLLM-DRIVEN ANALYSIS AND REMOVAL

Given the complexity of the interaction-consistent object removal task, we adopt a task decomposition strategy to structure the problem into three sequential subtasks: instruction understanding and reasoning, object localization and segmentation, and image inpainting-based removal.

The MLLM-driven analysis and removal process begins with the MLLM interpreting the instruction to identify both target objects and associated elements in the input image. We use OpenAI GPT-4o as our MLLM Analyzer. To improve the quality and completeness of this reasoning process, we do not directly prompt the MLLM to return a list of associated elements. Instead, we adopt a *chain-of-thought* prompting strategy (Wei et al. (2023)) that encourages the MLLM to explicitly reason through the implications of removing the target object. MLLM first describes the interaction dependencies and assesses whether each element should be removed. Only afterward are the mentioned elements collected and consolidated into a final removal list. This structured reasoning improves reliability, and ensures that subtle or indirect associations are more likely to be captured. For full prompts, please see Appendix B

Given the list of target objects and associated elements identified by the MLLM Analyzer, we then employ an open-vocabulary segmentation model, Grounded SAM (Ren et al. (2024)), to generate a binary mask for each identified element. A key consideration in object removal is to preserve

the original background as much as possible. By explicitly specifying the regions to be removed through accurate masks, we ensure that subsequent editing operations focus on the targeted areas. This localized control helps minimize unintended changes in unrelated regions.

The final step uses a mask-guided object removal model, ObjectClear (Zhao et al. (2025)), which takes the generated masks and removes all corresponding elements from the image. Since the masks include both target objects and associated elements, and the removal is guided by context-aware inpainting based on diffusion models, the resulting image tends to preserve global visual and semantic consistency. Diffusion models offer stronger generative capabilities, enabling plausible synthesis even in complex scenes.

This modular framework, which consists of an MLLM analyzer, an open-vocabulary segmentation model, and a mask-based object removal model, enables strong generalization across a wide range of real-world scenes. Each component can be flexibly replaced with improved alternatives. The framework is designed to operate entirely without additional training, while adapting effectively to diverse instructions and object relationships. The MLLM-driven design supports strong interpretability, as intermediate outputs at each stage can be easily examined to identify potential failure points.

## 4.2 MLLM-controlled Self-Correction

Diffusion models used in our mask-guided object removal may introduce hallucinated content or leave residual artifacts (Aithal et al. (2024), Cao et al. (2025)). Inspired by the concept proposed in Wu et al. (2024), we introduce an MLLM-controlled self-correction mechanism, which verifies and refines the object removal output through additional rounds of reasoning performed by the MLLM.

After MLLM Analyzer generates the removal target list including target objects and associated elements, an MLLM-based Simulator takes the input image and the removal list as input, and generates a textual description of the expected post-removal scene. This description serves as a semantic reference. Once the edited image is produced, an MLLM Examiner compares it against the simulated description to identify any remaining or inconsistent elements. These are compiled into a correction list. Both our MLLM Simulator and MLLM Examiner are implemented using GPT-4o.

Each correction item is then masked using an open-vocabulary segmentation model. To refine the output, a second round of mask-guided object removal with Attentive Eraser (Sun et al. (2025)) is applied. Unlike removal models used in Section 4.1, which emphasizes semantic-based object removal, this inpainting method fill masked regions by redirecting attention toward the surrounding context, ensuring that the newly identified elements are removed.

Our MLLM-controlled Self-Correction leverages the reasoning ability of MLLMs to perform a single-pass verification of the output, detecting and correcting any remaining inconsistencies or artifacts. This mechanism is especially beneficial when paired with weaker inpainting models, as it compensates for their limitations by improving consistency and artifact removal.

## 4.3 MLLM–LLM Collaboration for Local Deployment

**In practical deployments, object removal systems are often required to run on local devices due to privacy concerns, cost, reproducibility, or long-term accessibility of cloud-based APIs.** However, limited hardware resources on local devices restrict most users from running large-scale models, particularly LLMs, which demand substantial memory and computation for advanced reasoning. To address this, we develop a local variant of our framework that replaces large proprietary models with smaller open-source alternatives and is designed to operate efficiently on a single GPU.

One of the key challenges in this setting lies in the reduced reasoning capabilities of smaller MLLMs. In our experiments, compact MLLMs struggle to reliably act as MLLM Analyzer (Section 4.1). To mitigate this limitation, we adopt a *prompt chaining strategy* (Wu et al. (2022)) that decomposes the reasoning process into multiple substeps. Instead of asking the model to perform all reasoning in one pass, each step focuses on a simpler task, and the output from one step is passed as input to the next. This reduces the reasoning complexity at each stage and improves reliability under constrained model capacity.

Furthermore, we leverage the complementary strengths of different model types by employing a hybrid MLLM and LLM system. Since MLLMs allocate a significant portion of their parameters to

learning visual representations, their performance on commonsense reasoning tasks tends to lag behind that of text-only LLMs of similar size. To address this limitation, we assign image-related questions to the MLLM. On the other hand, purely textual reasoning is handled by a compact text-only LLM. This collaborative architecture allows us to preserve the semantic depth of reasoning required for ICOR, while maintaining compatibility with limited computational resources. The effectiveness of this design is validated in Section 5.3. Specifically, we separately deploy the MLLM llava-hf/llava-v1.6-vicuna-13b-hf (Liu et al. (2024)), quantized to 4-bit, and the LLM meta-llama/Llama-3.1-8B-Instruct (Grattafiori et al. (2024)), quantized to 8-bit.

As shown in Fig. 3, we break down the analysis phase into a series of sequential steps. First, LLM processes the instruction to identify the primary object that should be removed. Then, MLLM examines the input image and enumerates all associated elements. Based on this set, the LLM performs reasoning to imagine the scene in the absence of target objects, and determines which elements would become inconsistent and therefore should also be removed. Finally, the LLM consolidates target objects and associated elements into a refined and structured removal list, which is then passed to the segmentation model for mask generation. The self-correction module is omitted in the local variant for the sake of simplicity and efficiency. The entire framework runs efficiently on a single 24 GB GPU.

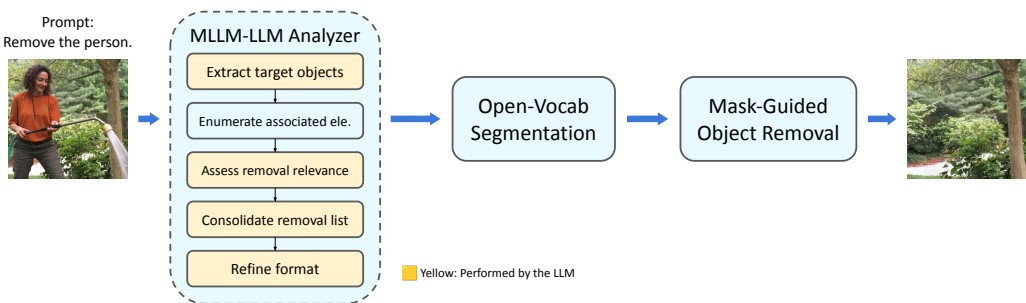

Figure 3: Overview of REORM for local deployment, which adopts prompt chaining strategy and coordinates a compact MLLM with an LLM.

## 5 EXPERIMENTS

### 5.1 EXPERIMENTAL SETUP

**Evaluation Dataset.** Existing object removal and image editing datasets provide valuable benchmarks but are not designed to evaluate models under interaction-consistent conditions. To fill this gap, we introduce **ICOREval**, a new benchmark dataset for assessing object removal in scenarios where the target object interacts contextually or physically with its surroundings.

ICOREval includes **110** examples, each with an input image containing objects involved in interactions, a natural language instruction specifying the object to be removed, and a ground-truth output image where the object is removed while preserving scene consistency. Instructions are manually written to reflect realistic user intents and do not describe interaction details, requiring models to infer the necessary contextual adjustments during removal.

The dataset is constructed from three sources: (1) Selected interaction cases from public datasets such as RORD (Sagong et al. (2022)), which records real-world scenes as fixed-view videos with objects added or removed across takes, and RemovalBench (Wei et al. (2025)), which captures the same scene twice (with and without the object) using a fixed camera; (2) Synthetic images generated with **object insertion models using Tewel et al. (2025) and Google Nano Banana**, where clean background images were generated and subsequently augmented by inserting objects in a contextually coherent manner; and (3) Copy-and-paste compositions, in which the target objects and their associated elements were manually pasted onto real-world background images to simulate interaction scenarios.

By combining real and synthetic data, ICOREval offers a compact yet challenging benchmark that evaluates not only the visual quality of the removal but also a model's ability to interpret instructions and reason about object–scene interactions.

**Evaluation Metrics.** We adopt four widely used metrics to evaluate object removal results: DINO score and LPIPS for perceptual similarity, and PSNR and SSIM for pixel-level fidelity. These provide a comprehensive assessment of both semantic consistency and reconstruction quality.

**Baselines.** We evaluate our method against several state-of-the-art baselines for language-guided image editing. The comparison includes MGIE (Fu et al. (2024)) and SmartEdit (Huang et al. (2024)), which employ MLLMs to interpret user instructions and guide the editing process. Both are open-source and can be run locally. In addition, we consider two proprietary models, OpenAI GPT Image 1[1] and Google Nano Banana[2], which support instruction-based image editing through commercial APIs. These closed-source models represent the current state of the art and serve as strong baselines for comparison.

## 5.2 COMPARISON WITH LANGUAGE-GUIDED IMAGE EDITING METHODS

**Quantitative Results.** Table 1 lists the quantitative comparison on ICOREval across all evaluated methods. Both variants of our proposed method, REORM (GPT-4o) and REORM (LLaVA+Llama), outperform all baselines across the evaluation metrics. Notably, REORM surpasses proprietary commercial systems, GPT Image 1 and Nano Banana, demonstrating its strong reasoning capabilities for achieving interaction-consistent object removal.

**We further evaluate the inference runtime of all compared methods. As shown in Table 1, our approach exhibits a longer inference time compared to other methods except for GPT Image 1. This is primarily attributed to the iterative reasoning process and the self-correction mechanism. Although these designs increase runtime, they substantially enhance the interaction consistency and visual fidelity of the final results, allowing REORM to outperform existing methods.** [3]

| Method | DINO ↑ | LPIPS ↓ | PSNR ↑ | SSIM ↑ | Runtime (s/img) |
|---|---|---|---|---|---|
| MGIE* | 0.709 | 0.261 | 19.217 | 0.634 | 5.20 |
| SmartEdit* | 0.760 | 0.275 | 20.064 | 0.615 | 10.09 |
| GPT Image 1[†] | 0.789 | 0.347 | 18.352 | 0.535 | 21.12 (API) |
| Nano Banana[†] | 0.873 | 0.124 | 25.225 | 0.771 | 8.45 (API) |
| Ours (LLaVA + Llama) | 0.897 | 0.121 | 25.674 | 0.820 | 12.54 |
| Ours (GPT-4o) | **0.937** | **0.104** | **27.063** | **0.825** | 13.15(API) + 4.33 |

Table 1: **Quantitative comparison** on ICOREval. Our REORM achieves the best performance across all four evaluation metrics (DINO, LPIPS, PSNR, and SSIM), while requiring a longer runtime. *Open-source methods that run locally. [†]Closed-source methods.

**Qualitative Results.** Fig. 4 presents qualitative comparisons on representative examples from ICOREval, illustrating the effectiveness of our method in achieving interaction-consistent object removal. (1) In the first example, REORM correctly infers that the person is riding a bicycle and removes both the rider and the bicycle, avoiding the physically implausible configuration of leaving the bicycle behind. (2) In the second example, our method removes the person together with their cast shadow. (3) In the third example, both variants of our model remove the person along with the (physically connected) watering can being held. (4) In the fourth example, our method removes a cat together with the toy it is playing with, including the attached string.

To further evaluate REORM's capability in ICOR, we collected a set of real-world cases from the web, covering a wider range of interaction scenarios. Fig. 5 presents the comparison results. (1) In the first example, both REORM variants successfully remove the target person along with their

---

[1]https://platform.openai.com/docs/guides/image-generation?image-generation-model=gpt-image-1

[2]https://ai.google.dev/gemini-api/docs/image-generation

[3]Methods that rely on remote inference through APIs are annotated accordingly. The runtime of GPT Image 1 and Nano Banana is reported as the average over three runs. All values are measured in seconds per image.

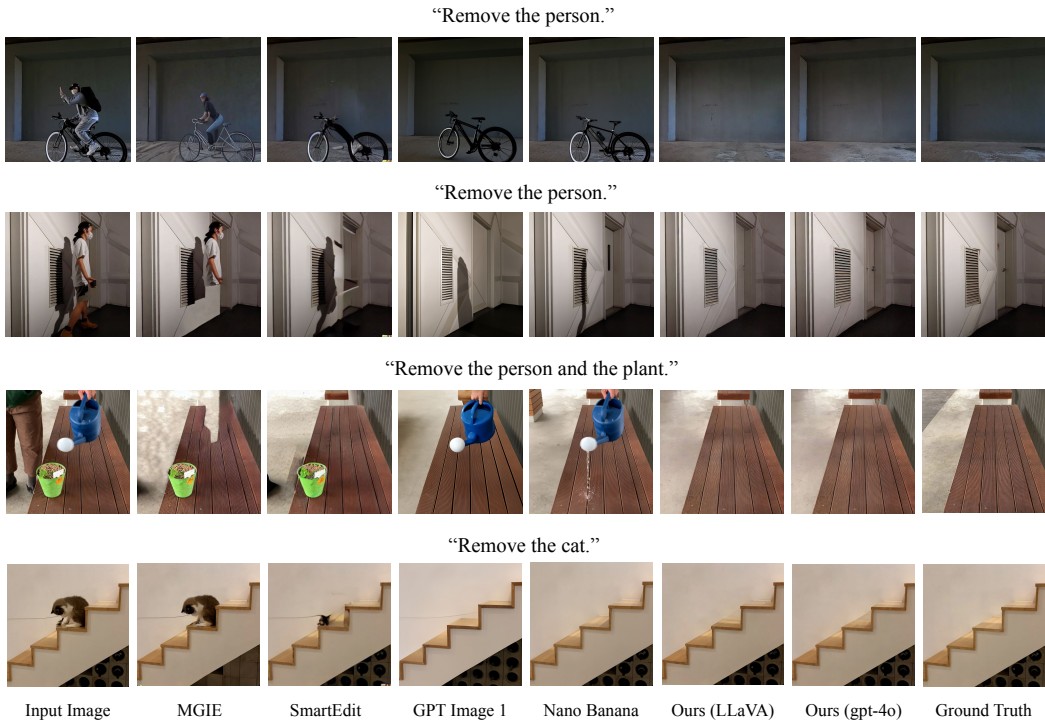

"Remove the person."

"Remove the person."

"Remove the person and the plant."

"Remove the cat."

| Input Image | MGIE | SmartEdit | GPT Image 1 | Nano Banana | Ours (LLaVA) | Ours (gpt-4o) | Ground Truth |

Figure 4: Qualitative comparison on ICOREval. Our method achieves ICOR, successfully removing a rider with the bicycle, a person with a cast shadow, a person holding a watering can, and a cat playing with a toy and its string.

cast shadow (lighting-dependent) and footprints (target-produced). (2) In the second example, after removing the hand, our method correctly infers that the physically connected watering can would appear to float and also removes the water stream being poured (target-produced). (3) The third example shows both variants removing a fire extinguisher together with its associated signage (contextually linked). (4) In the fourth example, only REORM (GPT-4o) successfully removes both the extinguisher and its sign, suggesting that a stronger MLLM can enhance generalization for this task. Please see also additional results in Appendix A.

## 5.3 VERIFICATION OF LOCAL DEPLOYMENT DESIGN

To validate the effectiveness of the local deployment design proposed in Section 4.3, we conduct a study examining two key techniques: prompt chaining and MLLM–LLM collaboration. These techniques are introduced to mitigate the reasoning limitations of compact MLLMs when deployed on resource-constrained devices. Table 2 compares three experimental settings. In experiment (a), LLaVA alone processes the same prompt as GPT-4o, without any form of prompt decomposition. In experiment (b), prompt chaining is applied to decompose the reasoning process into multiple steps, but still relying solely on LLaVA to execute all sub-tasks. **Notably, we observe that the performance in (b) degrades compared to (a). This decline is primarily attributed to error accumulation: the compact MLLM lacks sufficient reasoning capability to handle the decomposed steps accurately, causing errors in stages to cascade through the chain.**

Finally, experiment (c) combines both prompt chaining and MLLM–LLM collaboration. **By offloading logical reasoning stages to an LLM with stronger reasoning capabilities, we reduce the probability of errors, effectively resolving the error accumulation issue.** The results demonstrate that this full design achieves substantial improvements across all metrics, confirming that both techniques are necessary to enable robust reasoning under constrained conditions.

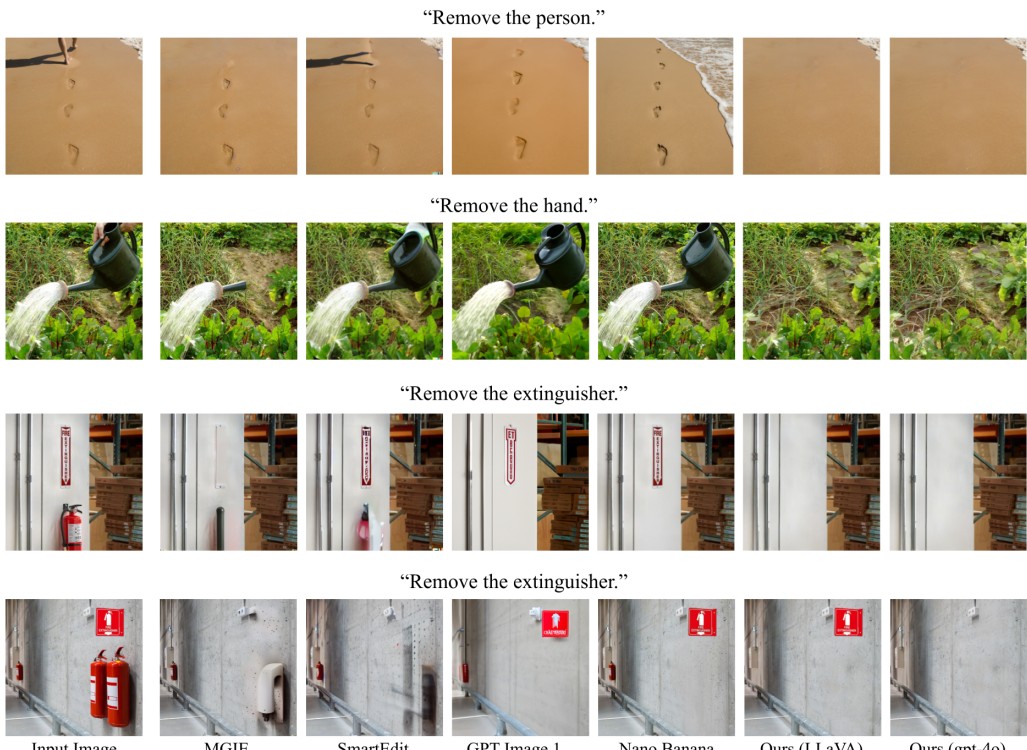

"Remove the person."

"Remove the hand."

"Remove the extinguisher."

"Remove the extinguisher."

| Input Image | MGIE | SmartEdit | GPT Image 1 | Nano Banana | Ours (LLaVA) | Ours (gpt-4o) |

Figure 5: Qualitative comparison on images without ground truth. REORM removes not only target objects but also associated elements such as shadows, footprints, watering can and streams, and signage, with the GPT-4o variant showing stronger generalization in challenging cases.

| Exp. | MLLM | Prompt Chaining | LLM | DINO ↑ | LPIPS ↓ | PSNR ↑ | SSIM ↑ |
|------|------|-----------------|-----|--------|---------|--------|--------|
| (a) | ✓ | | | 0.921 | 0.125 | 25.266 | 0.810 |
| (b) | ✓ | ✓ | | 0.913 | 0.128 | 25.230 | 0.813 |
| (c) | ✓ | ✓ | ✓ | **0.943** | **0.109** | **26.033** | **0.830** |

Table 2: **Verification of local deployment design** on ICOREval. **This experiment was conducted on the initial version of ICOREval (72 examples).** Experiment (a) uses LLaVA alone; (b) applies prompt chaining with LLaVA only; and (c) combines prompt chaining with MLLM–LLM collaboration. The full design in (c) shows improved performance across all metrics, indicating that both techniques are necessary for robust reasoning under constrained conditions.

## 5.4 FAILURE CASE ANALYSIS

As illustrated in Fig. 6, our method occasionally inadvertently removes background elements that should be preserved. For instance, the black object in the background (left) and the decorative items on the sofa (right) are incorrectly erased. This phenomenon stems from the self-correction mechanism, which compares the initial result with the expected post-removal scene description to trigger a second round of removal. However, the generated description occasionally fails to capture every detail of the background objects. When a background element is omitted from the scene description, the system interprets its presence as a residue to be removed, leading to false positives and subsequent over-editing. In the right case, for instance, the expected post-removal description does not mention the conical ornament, leading the MLLM Examiner to incorrectly treat it as an object that should be removed.

It is worth noting that the number of errors rectified by the self-correction module outweighs these newly introduced artifacts. Nevertheless, mitigating such false positives and further reducing over-editing remains a valuable direction for future research.

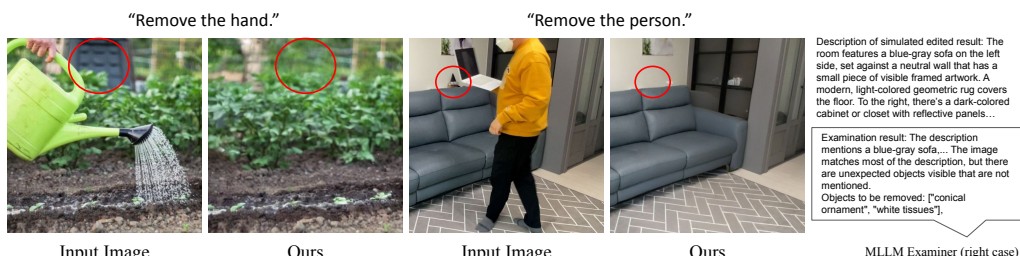

Figure 6: **Failure cases. Our method occasionally incorrectly removes background elements. The rightmost column displays the description of simulated edited result (input to the MLLM Examiner), while the speech bubble contains the corresponding output.**

## 5.5 Discussion: Toward Intent-Aware and Privacy-Preserving Object Removal

While interaction-consistent object removal (ICOR) aims to generate semantically consistent and visually realistic results by removing not only the target object but also its associated interaction effects, this objective may not always align with the user's editing intent. In some scenarios, users may prefer to retain certain contextual elements—for instance, when creating special effects, producing intentionally unrealistic images, or making minimal edits limited to the target object.

In such cases, enforcing interaction-aware removal can lead to over-editing or unintended modifications that conflict with user preferences. This underscores the importance of considering user intent when executing editing instructions. Ideally, an object removal or image editing system should adapt to diverse intents that may vary across tasks, image content, or individual editing habits. However, implicitly inferring user intent without additional input remains a challenging problem. Another emerging concern is privacy, which has drawn increasing attention in real-world deployment. Our local deployment design also represents an initial step toward addressing this challenge. Together, these issues point to future research directions in developing more intelligent image editing systems that balance user autonomy, efficiency (through intent prediction and personalization), and privacy protection under resource-constrained conditions.

## 6 Conclusion

We introduce Interaction-Consistent Object Removal, which extends object removal by jointly removing interaction elements that would otherwise compromise scene plausibility. To address this task, we propose Reasoning-Enhanced Object Removal with MLLM (REORM), a framework that leverages the reasoning capabilities of MLLMs to identify related elements for removal. We constructed ICOREval to evaluate this task and demonstrated that REORM outperforms existing image editing models. We believe that the formulation of interaction-consistent object removal opens up new directions for object removal and image editing.

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

# A    APPENDIX: ADDITIONAL RESULTS

Fig. 7 presents additional qualitative results on the ICOREval benchmark. (1) In the first example, our method removes both the person and their reflection in the glass, whereas Nano Banana fails to remove the reflection and GPT Image 1 introduces noticeable background changes affecting the floor and wall appearance. (2) In the second example, GPT Image 1 produces an artifact in the handrail region, while Nano Banana fails to remove the person. (3) In the third example, ours (LLaVA) does not fully remove the person's arm, but this limitation is corrected by our self-correction mechanism, as shown in ours (GPT-4o), which generates a more complete and consistent result.  (4) In the fourth example, GPT Image 1 unnecessarily alters the person's appearance (removing the hat), while Nano Banana and ours both preserve the identity and successfully remove the dog and leash. (5) In the fifth example, GPT Image 1 adds and relocates boxes onto the floor. Although not visually implausible, this highlights the importance of user intent prediction, as discussed in Section 5.5. Subtle background changes are also observed in the floor carpet and sofa pillows. (6) Finally, in the last example, GPT Image 1 slightly alters the tile near the foreground, which becomes noticeable only upon close inspection compared to Nano Banana, ours, and the ground truth.

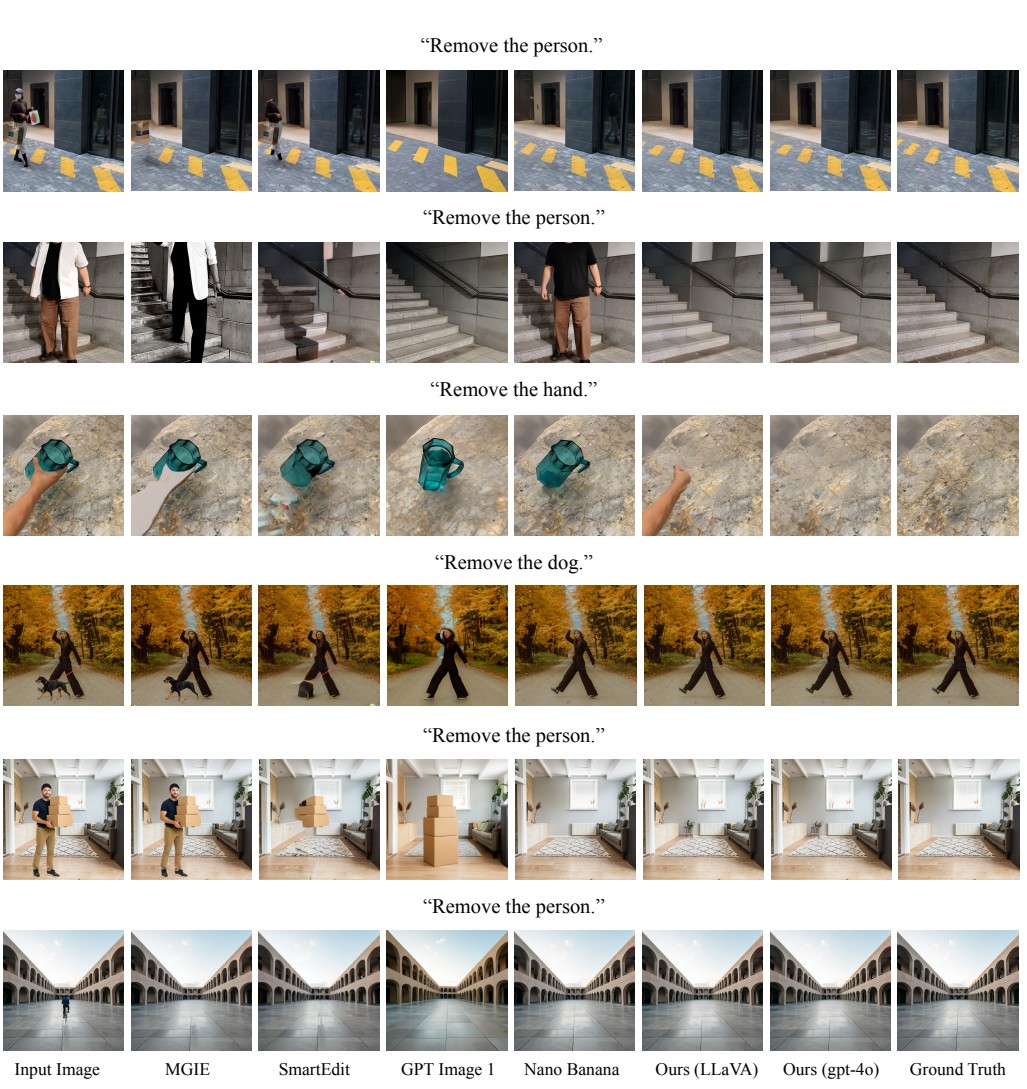

Figure 7: Qualitative comparison of different methods on ICOREval.

Figure 8 shows additional qualitative comparisons on images without ground truth annotations. (1) In the first example, only our method successfully removes both the person and their footprints. (2) In the second example, our method is the only one that removes both the trash bin and its associated signage, while other methods either leave the sign or remove only part of the bin. (3) In the third example, the result of GPT Image 1 leaves the garden hose that appears to float unnaturally. (4) In the fourth example, none of the other methods successfully remove the cat toy. Moreover, Nano Banana alters the appearance of the toy. (5) In the fifth example, GPT Image 1 leaves behind an implausible water stream, while Nano Banana fails to remove the watering can and leaves it unnaturally suspended in the air. (6) In the sixth example, ours (LLaVA) does not remove the dog's shadow, but ours (GPT-4o) produces a consistent and complete result, demonstrating the benefit of stronger reasoning capabilities.

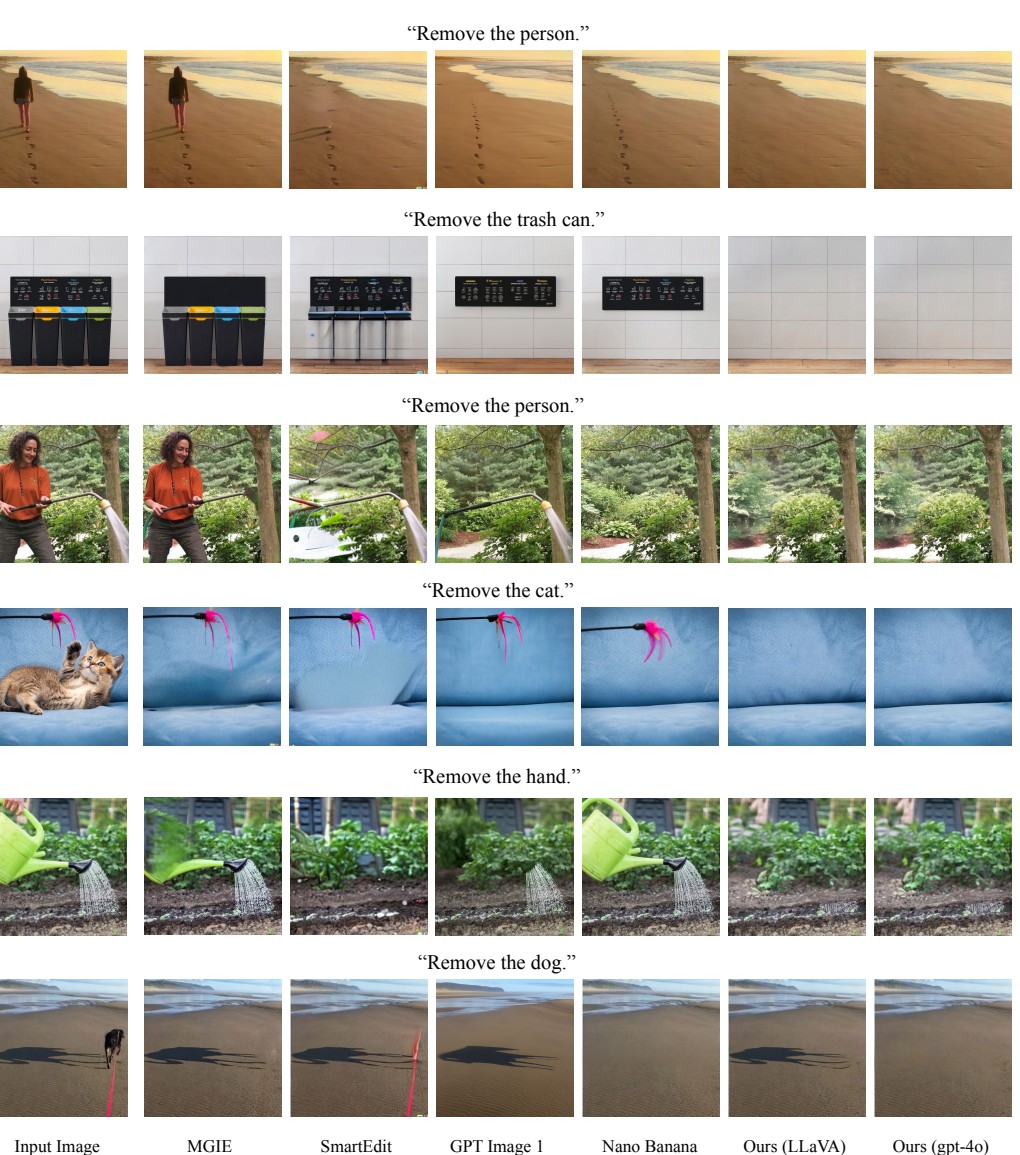

Figure 8: Qualitative comparison of different methods on images without ground truth.

# B  APPENDIX: PROMPTS FOR THE MLLMS USED IN OUR FRAMEWORK

## B.1  MLLM ANALYZER

```
# Identity
Helpful assistant that can help the user to identify the target objects and associated
elements that should be removed from the image, ensuring the final result is contextually
coherent.

# Your task
Given an image and a prompt, you need to analyze and identify the target objects and all
related objects and effects that should be removed in the image.  You need to provide a
list of objects that should be removed in the image.

# Process Steps
1.  Read the user prompt and identify the target objects with the attributes.
2.  Carefully examine the image to identify any other objects or effects whose presence
would become physically implausible or contextually illogical if the primary target object
were removed.  This includes, but is not limited to:
  * Lighting-dependent effects:  shadows, reflections.
  * Physically connected objects:  items being held, worn, or physically supported by the
target.
  * Target-produced elements:  effects or objects created by the target, such as water from
a sprinkler.
  * Contextually linked objects:  items that lose their purpose or context without the
target, such as a sign for an object, a mailbox for a house, or a leash for a dog.
3.  Explain your understanding and then format your answer as shown in the examples.
4.  Avoid vague or abstract noun phrases followed by relative clauses, such as ``the
object that...''  or ``any item that...''.  Use specific object names only.  For example,
write ``the dog's toy'' instead of ``the object it is interacting with''.
5.  Avoid describing subtle background changes or textures.

# Example
<product_review id="example-1">
Remove the person.
</product_review>
<assistant_response id="example-1">
Reasoning:  ``The target object is the person.  If the person is removed, his shadow and
his scooter would appear contextually inconsistent, as the scooter would appear to stand
upright without a rider.''
Target Objects:  ["person", "the person's shadow", "the scooter"]
</assistant_response>

<product_review id="example-2">
Remove the white dog.
</product_review>
<assistant_response id="example-2">
Reasoning:  ``The target object is a white dog.  Removing the dog would make its
reflection in the water and the toy it is playing with appear contextually inconsistent,
as the toy would not reasonably be there without the dog.''
Target Objects:  ["white dog", "the white dog's reflection", "the dog toy"]
</assistant_response>

<product_review id="example-3">
Remove the person.
</product_review>
<assistant_response id="example-3">
Reasoning:  ``The target object is a person.  Without the person, the bags and the cup he
is holding would appear to float in midair, which is physically implausible.''
Target Objects:  ["person", "the bags", "the cup"]
</assistant_response>
```

Table 3: Our full prompt for the MLLM Analyzer.

## B.2 MLLM Simulator

```
# Identity
You are a vision-language assistant that describes how the picture will look after the
requested objects are removed.

# Your task
Given one image and the user's removal request, describe the resulting image.

# Output Rules
1.  Describe all visible elements in the resulting image as detailed as possible.  Include
all elements in the background, foreground, and any other visible parts of the image.
2.  Avoid abstract emotions or psychology.
3.  Do NOT mention the original image, the removal process, or phrases like ``in the given
image.''
```

Table 4: Our full prompt for the MLLM Simulator.

## B.3 MLLM Examiner

```
# Identity
You are a vision-language assistant that checks whether an edited image matches the
description.

# Your task
Given one image and a description, examine whether the image contains any unexpected
objects or elements not mentioned in the description.  If there are such elements, list
them.  Otherwise return an empty list.

# Output Rules
1.  Provide the reasoning first, and then list the objects that should be removed from the
image.
2.  Only include objects that should be removed.
3.  Ignore discrepancies in the quantity of objects.
4.  Only include the specific name of the object, not its location.  State the name
directly (e.g., ``a blue vase''), not in a vague way (e.g., ``an object that looks like
a vase'').  Include the color or shape if you are certain about it.
5.  If you are uncertain about an object's precise identity, do not list it.
6.  Avoid using vague or large-scale location words such as ``area'', ``space'', or
``region''.  Focus only on specific, tangible objects.

# Examples
<product_review id="example-1">
The edited image features a kitchen with a stove, a sink, and a refrigerator.  The stove
is clean and shiny, the sink is empty, and the refrigerator is closed.  There are no
people or pets in the image.
</product_review>
<assistant_response id="example-1">
Reasoning:  ``The image shows a cozy kitchen.  It contains a stove, a chair, and a
refrigerator.  The chair is not mentioned in the description, so it should be removed.
There's something like a hand in the image, which doesn't appear in the description.''
Objects to be removed:  ["the chair", "the hand", "arm"]
</assistant_response>
```

Table 5: Our full prompt for the MLLM Examiner.

# C  APPENDIX: DATASET DISTRIBUTION AND DIVERSITY

To provide a comprehensive understanding of the dataset's composition and diversity, we present a statistical breakdown in Table C. Regarding the Associated Elements Categories shown on the left, we categorize the elements into four types as defined in Section 3. It is important to note that these categories are not mutually exclusive; a single image frequently contains multiple associated elements simultaneously, which increases the complexity and realism of the evaluation. Regarding the Data Construction Source on the right, images are collected via three distinct methods, as introduced in Section 5.1.

| Associated Elements Categories | | Data Construction Source | |
| --- | --- | --- | --- |
| Type | Count | Type | Count |
| Lighting-dependent | 53 | Public datasets | 46 |
| Physically connected | 35 | Synthetic images | 31 |
| Target produced | 16 | Copy-and-paste | 33 |
| Contextually linked | 28 | – | – |

Table 6: Dataset distribution statistics. The left side details the interaction categories, and the right side shows the data collection sources. Note that a single image may contain multiple associated elements simultaneously.

To assess the visual diversity and semantic coverage of ICOREval, we compare its feature distribution against three baselines: MagicBrush Zhang et al. (2023), RemovalBench Wei et al. (2025), and OBER-Test Zhao et al. (2025). We employ the pre-trained CLIP (ViT-B/32) model to map images into a high-dimensional semantic space. For each image, we extract the global embedding and apply $L_2$ normalization to ensure the analysis focuses on semantic directionality. To facilitate a fair and unbiased density comparison, we randomly sample a subset from the larger dataset to match the number of samples in the smaller dataset ($N = \min(N_{ours}, N_{baseline})$). These high-dimensional embeddings are then projected into a two-dimensional manifold using t-SNE with PCA initialization to preserve global structures.

The resulting visualization, presented in Figure 9, situates ICOREval (blue) within the semantic landscape of existing benchmarks. While the open-domain MagicBrush exhibits the widest distribution, ICOREval achieves a competitive level of diversity. Crucially, ICOREval outperforms object removal benchmarks; as shown in the center and right plots, OBER-Test and RemovalBench (gray) tend to cluster in limited sub-regions, whereas ICOREval maintains a wider distribution. This indicates that our dataset offers great semantic coverage compared to other baselines.

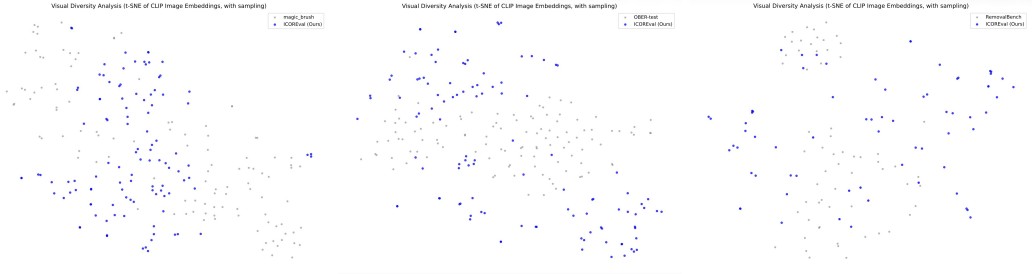

Figure 9: Visual diversity analysis using t-SNE on CLIP embeddings. We compare ICOREval (Blue) against MagicBrush, OBER-Test, and RemovalBench (Gray). The visualization shows that ICOREval maintains a broad semantic distribution that is more diverse than OBER-Test and RemovalBench, though less extensive than the open-domain MagicBrush.

To further quantify the semantic complexity of ICOREval beyond 2D visualization, we examine its intrinsic dimensionality using Principal Component Analysis (PCA) on the 512-dimensional CLIP

embeddings. We analyze the cumulative explained variance ratio to determine the number of principal components required to represent the dataset's information. Figure 10 reveals that the top 5 principal components account for only 32.44% of the total variance, indicating that the data distribution is not dominated by a few primary features. Furthermore, 52 and 67 principal components are required to capture 90% and 95% of the total variance, respectively. This requirement for a significant number of dimensions suggests that ICOREval possesses a rich semantic structure with diverse visual attributes, rather than being confined to a low-dimensional manifold.

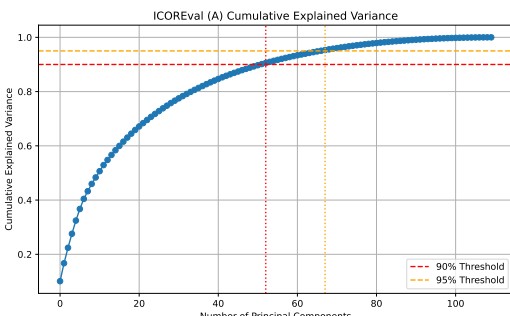

Figure 10: Intrinsic dimensionality analysis of ICOREval via PCA on CLIP embeddings. The curve illustrates the cumulative explained variance ratio. The vertical dashed lines indicate that 52 and 67 principal components are required to capture 90% (red) and 95% (orange) of the total variance, respectively.

