# OpenReview forum: "Interaction-Consistent Object Removal via MLLM-Based Reasoning"
_ICLR.cc/2026/Conference — Submitted to ICLR 2026_

### Official Review · Reviewer_PobY · 2025-10-22

**Soundness:** 2
**Presentation:** 2
**Contribution:** 2
**Rating:** 4
**Confidence:** 3

**Summary:**

The paper proposes REORM, a system for interaction-consistent object removal (ICOR). It combines a multimodal large language model (MLLM) for reasoning about which related elements should be removed, with a mask-guided inpainting module, a self-correction mechanism, and a lightweight local variant. The authors also introduce a new evaluation benchmark, ICOREval, and report better quantitative results than existing image-editing methods.

**Strengths:**

The paper clearly formulates a refined variant of the object removal task that accounts for both the target object and its surrounding contextual or interacting elements, such as shadows, reflections, and connected parts. This makes the problem definition more realistic and relevant to real-world editing scenarios. The authors build a complete end-to-end system and introduce a new benchmark, ICOREval, specifically designed to evaluate interaction-consistent removal quality. Experimental results indicate moderate but consistent improvements over strong baselines, and the visual examples show more coherent and natural editing outcomes in several challenging cases.

**Weaknesses:**

1. There seems to be a collateral deletion issue — for example, in Fig. 1 (iii), background objects are also removed, indicating the presence of erroneous behavior.

2. The error analysis is insufficient. It would be valuable to discuss whether error accumulation occurs when combining LLMs and MLLMs.

3. The method has not been evaluated on existing benchmark datasets, such as MagicBrush, which limits the fairness and completeness of the comparison.

4. It is recommended to include inversion-based image editing and inpainting methods in the comparison or discussion.

5. The proposed approach appears to be computationally expensive; more experiments or analyses on computational cost should be provided.

6. Given that introducing LLMs and MLLMs may enable more accurate mask extraction, traditional inpainting methods could potentially achieve strong results as well.

7.Overall, the proposed method lacks novelty. It does not clearly define a new problem and the technical contributions mainly rely on modular integration rather than introducing fundamentally new ideas.

**Questions:**

Same as weaknesses

---

> ### Author Response · Authors · 2025-11-23
> **Our Reply to Reviewer PobY (W1 to W6)**
>
> We thank the reviewer for providing the feedback. In the following, we respond to each of the concerns in the review. For clarity of reference throughout this response, we denote the specific Weaknesses as W1, W2, etc.
>
> 1. **Failure Cases** (W1)
> We thank the reviewer for this insightful observation regarding the collateral deletion. We acknowledge that this is an instance of over-editing caused by our self-correction mechanism. We have included a discussion on such failure cases in Section 5.4 and Figure 6 in the revised PDF. Although this mechanism can inadvertently cause over-editing in rare cases, it ensures cleaner removals for the majority of samples, making it a favorable trade-off for robust removal.
>
> 2. **Does error accumulation occur when combining LLMs and MLLMs** (W2)
> We thank the reviewer for raising this critical question. We have updated Section 5.3 to explicitly address this phenomenon. Our ablation study demonstrates that while error accumulation indeed occurs when using prompt chaining with a compact MLLM alone (Experiment b), our proposed MLLM-LLM collaboration (Experiment c) effectively mitigates it. By offloading complex logic to an LLM with stronger reasoning capabilities, we significantly reduce initial errors, thereby preventing the error cascade observed in single-model setups.
>
> 3. **The method has not been evaluated on existing benchmark datasets** (W3)
> We understand the reviewer's concern regarding not evaluated on existing datasets. However, we respectfully emphasize that Interaction-Consistent Object Removal is a newly defined task that is not addressed by existing benchmark datasets like MagicBrush.
> The primary limitation of existing datasets is the lack of interaction relationships (or associated elements). These datasets typically focus on general image editing or single-object removal, without the physical and interaction dependencies between objects. This fundamental gap in existing data is precisely the motivation behind the creation of our own dedicated benchmark, ICOREval, which explicitly contains these complex interaction scenarios.
>
> 4. **Recommended to include inversion-based image editing and inpainting methods in the comparison or discussion** (W4)
> Thank you for the suggestion. Inversion-based editing methods, such as PnP Inversion [ICLR’24], SDEdit [ICLR’22], and Prompt-to-Prompt editing [ICLR’23], tend to be computationally expensive and provide limited flexibility for structural modifications, including adding or removing objects [as noted in BrushEdit].
>
> 5. **Runtime analysis for the overall pipeline** (W5)
> We appreciate the reviewer's query regarding the computational efficiency and resource requirements of our REORM framework. We have included a runtime analysis in the updated PDF (Section 5.2), and the following addresses your question directly:
> As shown in Table 1 (`Runtime' column), our approach exhibits a longer inference time compared to most other methods except for GPT Image 1. This is primarily attributed to the sophisticated and necessary reasoning process within our MLLMs, which is central to achieving high interaction consistency. The increased runtime stems from three key components:
>     * (1) The MLLM reason from the image to correctly identify all associated elements that should be jointly removed.
>     * (2) Ours(GPT-4o) incorporates an self-correction mechanism.
>     * (3) Multiple rounds of inference are performed by Ours(LLaVA + Llama).
>
>     While this sophisticated, reasoning-based process necessitates a longer runtime, this is a conscious trade-off for achieving substantial gains in the ability to handle complex removal sequences requiring deep semantic understanding. Our superior performance in interaction consistency justifies the increased computational overhead.
>
> 6. **With LLMs'/MLLMs' mask extraction ability, traditional inpainting methods could potentially achieve strong results as well** (W6)
> Thank you for the comment. Our goal is to enable object removal systems to handle interaction-related elements, which requires higher-level scene reasoning rather than mask extraction alone. Although traditional inpainting methods can benefit from improved masks, they are not sufficient to determine which dependent elements should be jointly removed. Our framework shows how LLMs and MLLMs can be integrated with inpainting methods to support this reasoning process and achieve interaction-consistent removal. Without such reasoning-guided integration, the ICOR objective cannot be reliably achieved.

---

> > ### Author Response · Authors · 2025-11-23
> > **Our Reply to Reviewer PobY (W7)**
> >
> > 7. **The proposed method lacks novelty** (W7)
> > We appreciate this opportunity to clarify our contributions. While our framework leverages existing tools, we respectfully posit that the core innovation lies not merely in the integration, but in defining a novel problem space and introducing specific reasoning mechanisms.
> >     * **Novel Task Definition and Taxonomy:**
> > Our primary contribution is the formalization of the "Interaction-Consistent Object Removal" task. Unlike general inpainting, this task requires resolving interaction dependencies between objects. We not only define this new problem but also systematically categorize it into four interaction types, establishing a clear taxonomy that makes this complex problem addressable and scalable for future research.
> >     * **Methodological Innovations:**
> > REORM is the first baseline specifically designed to tackle the new task. Beyond the pipeline structure, we introduce two specific algorithmic innovations:
> > **Self-Correction Mechanism**: We introduce a verification mechanism where the MLLM generates a textual description of an "ideal post-edit state" to serve as a ground truth for evaluating the preliminary visual result. Leveraging the MLLM's extensive semantic priors is a novel approach to ensuring semantic fidelity.
> > **MLLM-LLM Collaboration**: For our local variant, we design a unique collaboration between MLLMs and LLMs. We decouple visual perception (handled by MLLM) from complex logical reasoning (handled by LLM). By applying the complex text reasoning required for prompt chaining to the LLM, we achieve superior interaction reasoning within limited resource constraints.
> > Prior works such as BrushEdit [1] and Magicquill [2] don't possess these designs.

---

> > > ### Comment · Reviewer_PobY · 2025-11-28
> > > **Response to Authors**
> > >
> > > I would like to thank the authors for their response. I would prefer to see the evaluations of the other reviewers before making a final recommendation.

---

### Official Review · Reviewer_5SoM · 2025-10-26

**Soundness:** 2
**Presentation:** 3
**Contribution:** 2
**Rating:** 4
**Confidence:** 5

**Summary:**

This paper proposes a task called Interaction-Consistent Object Removal, which aims to remove not only the specified object but also other elements that interact with it in the scene. The framework uses multimodal large language models to infer and guide the removal of interacting objects. The framework includes an MLLM-driven reasoning module, a self-correction mechanism, and a locally deployable lightweight variant. A new dataset containing 72 examples is also introduced to evaluate semantic and visual consistency.

**Strengths:**

1. The paper introduces a meaningful task, Interaction-Consistent Object Removal, extending traditional object removal to model semantic and physical interactions among scene elements.
2. The inclusion of a local deployment variant demonstrates practical value and thoughtful consideration of real-world constraints.

**Weaknesses:**

1. The ICOR task can be viewed as an extension of prior object-effect removal methods, expanding from handling only lighting-dependent effects to covering more types, such as physically connected objects, target-produced elements, and contextually linked objects. However, if paired training data were available, prior methods might also be able to handle these limited cases, which would limit the novelty.
2. The ICOREval dataset, while a contribution of the paper, includes only 72 examples, which is limited and may introduce bias in evaluating task effectiveness.
3. The paper lacks discussion of failure or over-removal cases, and does not analyze when the reasoning process misfires or produces incorrect object associations.
4. Although intent-aware editing is mentioned in Section 5.4, no mechanism is proposed to adapt to different user intentions or control levels, leaving controllability unexplored.

**Questions:**

1. Issues raised in the *Weaknesses* section.
2. Are there plans to expand the ICOREval dataset for broader benchmarking?
3. Are there examples where self-correction fails or leads to over-removal?

---

> ### Author Response · Authors · 2025-11-23
> **Our Reply to Reviewer 5SoM (W1 to W4 and Q1 to Q3)**
>
> We thank the reviewer for providing the feedback. In the following, we respond to each of the concerns in the review. For clarity of reference throughout this response, we denote the specific Weaknesses as W1, W2, etc. and Questions as Q1, Q2, etc.
>
> 1. **If paired training data were available, prior methods might also be able to handle these limited cases, which would limit the novelty.** (W1)
> We appreciate the reviewer's suggestion. However, we would like to highlight the contribution of our work based on two key factors: the intrinsic value of our zero-shot framework and the fundamental difficulty of the supervised alternative.
>     * **Value of zero-shot and interpretability**:
> Our method is strategically positioned as a zero-shot solution, eliminating the need for resource-intensive training. Beyond efficiency, the modular nature of our MLLM-based framework offers a significant advantage in interpretability and trustworthiness—qualities often lacking in end-to-end trained black-box models. By leveraging MLLMs, our pipeline allows users to explicitly track and diagnose errors at each stage (reasoning, segmentation, or inpainting), as exemplified by the failure case analysis in Section 5.4. This transparency makes the system more robust and controllable in real-world applications.
>     * **The intractability of training-based approaches**:
> While training-based methods might work if data were available, the premise of acquiring such data is non-trivial. Constructing a large-scale dataset with complex interaction dependencies is exceptionally difficult. Unlike standard inpainting datasets, this task requires annotations that capture nuanced physical and semantic relationships between objects. Furthermore, to achieve good generalization, such a dataset would need to exhaustively cover all four interaction categories we defined. Our work bridges this gap precisely because it functions effectively without requiring this prohibitively expensive data curation.
>
> 2. **Expansion of ICOREval evaluation dataset** (W2, Q2)
> We appreciate the reviewer's constructive suggestion regarding the dataset size and acknowledge that a larger benchmark strengthens the statistical robustness of our conclusions.
> Taking immediate action, we have expanded ICOREval to 110 examples. For context, this scale compares favorably with or exceeds existing datasets in this domain, such as RemovalBench (70), ObjectDrop (100), and OBER-Test (160).
> We have re-evaluated our method on this expanded dataset, and the updated quantitative results are presented in Table 1 of the revised PDF. The results on the expanded set remain consistent with our initial findings, further validating the effectiveness of our approach.
>
> 3. **Discussion of failure or over-removal cases** (W3, Q3)
> We thank the reviewer for this valuable suggestion. We have included a discussion on failure cases in Section 5.4 and Figure 6 in the revised PDF. Although the self-correction mechanism can inadvertently cause over-editing in rare cases, it ensures cleaner removals for the majority of samples, making it a favorable trade-off for robust removal.
>
> 4. **No mechanism is proposed to adapt to different user intentions or control levels** (W4)
> We acknowledge the reviewer's valid observation regarding controllability. As discussed in Section 5.4, we consider intent-aware editing to be a broad and highly challenging research topic that warrants its own dedicated investigation. Therefore, we have positioned it as a direction for future work.
> To address this challenge, a user modeling mechanism to construct a user model that can better interpret a user's editing intent. We believe implicit user modeling during the interaction process is a particularly promising approach, for example, leveraging LLMs to ask clarifying questions for preference elicitation [1]. This would allow the system to dynamically adapt to different user intentions and control levels, significantly enhancing controllability.
> [1] Asking Clarifying Questions for Preference Elicitation With Large Language Models

---

> > ### Comment · Reviewer_5SoM · 2025-11-25
> >
> > Thank you for the detailed responses and the revisions. However, I remain concerned about the diversity and potential biases of the ICOREval dataset, especially given that the benchmark is positioned as a core contribution of the paper and may be used by the community in future evaluations. While I appreciate the effort required to expand such a dataset, it would strengthen the paper if the authors could provide an analysis of the diversity and a discussion of potential biases.

---

> > > ### Author Response · Authors · 2025-11-27
> > > **Reply to Reviewer 5SoM: diversity analysis of the ICOREval dataset**
> > >
> > > We sincerely thank the reviewer for highlighting this critical aspect. We've conducted a comprehensive analysis of the dataset’s distribution, semantic coverage, and intrinsic complexity. We've added these detailed results to **Appendix 3** of the revised paper. The new analysis provides quantitative and qualitative evidence of ICOREval’s diversity:
> > >
> > > 1. **Dataset distribution statistics (Table 6):** Table 6 presents the detailed composition of ICOREval, organized across two dimensions: Interaction Categories and Data Sources. The dataset covers four distinct interaction types, capturing the complex, overlapping nature of real-world associations. Additionally, we employed a hybrid construction strategy by collecting images from three different sources.
> > >
> > > 2. **Broad Semantic Coverage via t-SNE (Figure 9):** We performed a comparative analysis using t-SNE on CLIP (ViT-B/32) embeddings to visualize the semantic landscape of ICOREval against existing benchmarks. The result shows that ICOREval maintains a broad semantic distribution that is more diverse than OBER-Test and RemovalBench, though less extensive than the open-domain MagicBrush. This confirms that ICOREval covers a wide spectrum of visual scenarios and avoids the domain-narrowing bias often seen in specialized evaluation sets.
> > >
> > > 3. **High Intrinsic Dimensionality via PCA (Figure 10):** To quantitatively rule out redundancy, we analyzed the intrinsic dimensionality of the dataset using Principal Component Analysis (PCA). The results show that the top 5 principal components explain only 32.44% of the variance, and it requires 52 principal components to capture 90% of the total variance. This high dimensionality serves as strong mathematical evidence that the dataset possesses a rich semantic structure and is not dominated by a few repetitive features.
> > >
> > > We believe these additions provide a transparent and robust characterization of ICOREval, confirming its suitability as a core contribution to the community.

---

### Official Review · Reviewer_gsTw · 2025-11-01

**Soundness:** 3
**Presentation:** 3
**Contribution:** 2
**Rating:** 4
**Confidence:** 5

**Summary:**

This paper introduces Interaction-Consistent Object Removal (ICOR), a new formulation of the object-removal task that requires eliminating not only the target object but also its interaction-related elements—such as shadows, reflections, connected objects, or contextually dependent items—to maintain semantic and visual coherence.
The authors propose REORM, a modular reasoning-enhanced framework that leverages multimodal large language models (MLLMs) to infer what secondary elements must also be removed. The system combines MLLM-driven reasoning, open-vocabulary segmentation, and diffusion-based inpainting, followed by an MLLM-controlled self-correction stage.
A new benchmark, ICOREval, is constructed for evaluation. Experiments show REORM achieves better visual quality and semantic consistency than existing instruction-based image editors such as MGIE, SmartEdit, GPT-Image-1, and Nano Banana.

**Strengths:**

### Originality
Defines a new task: Interaction-Consistent Object Removal, emphasizing removal of both objects and their interaction elements. The introduction of Interaction-Consistent Object Removal extends classical object removal to a more semantically complete problem, emphasizing real-world plausibility.

### Quality
Technically solid modular design combining reasoning, segmentation, and diffusion-based inpainting.

### Clarity
The paper is clearly structured with well-designed figures explaining each module.

### Significance
The focus on removing interaction-linked elements (lighting effects, physical connections, contextual dependencies) is novel and practically valuable for photo editing and generative scene understanding.

**Weaknesses:**

1. While the task definition is new, the core framework mainly orchestrates existing tools (GPT-4o, Grounded-SAM, ObjectClear). The contribution lies more in integration and reasoning design than in fundamental algorithmic innovation.

2. The use of MLLM reasoning to guide inpainting is not entirely novel — similar techniques appear in BrushEdit[1] and Magicquill[2].

3. The main version depends on closed APIs (GPT-4o), raising concerns about reproducibility, cost, and long-term accessibility. The paper could discuss these trade-offs more explicitly.

4. ICOREval contains only 72 examples, which limits the statistical robustness of conclusions. Larger or more diverse benchmarks would strengthen generalizability claims.

5. The metric design focuses on perceptual similarity (DINO, LPIPS, SSIM) rather than explicitly measuring semantic consistency or over-editing, leaving the notion of interaction consistency somewhat subjective.

[1] Li Y, Bian Y, Ju X, et al. Brushedit: All-in-one image inpainting and editing[J]. arXiv preprint arXiv:2412.10316, 2024.
[2] Liu Z, Yu Y, Ouyang H, et al. Magicquill: An intelligent interactive image editing system[C]//Proceedings of the Computer Vision and Pattern Recognition Conference. 2025: 13072-13082.

**Questions:**

1. How is interaction consistency operationally defined or measured beyond qualitative examples? Could an explicit metric be introduced?
2. Beyond the engineering-level modular integration, what constitutes the paper’s core innovation? Similar pipeline connections have appeared in prior work, which somewhat limits the novelty of this contribution.
3. What is the runtime and resource requirement per image for the overall pipeline?

---

> ### Author Response · Authors · 2025-11-23
> **Our Reply to Reviewer gsTw (W1 to W4, Q2 to Q3)**
>
> We thank the reviewer for providing the feedback. In the following, we respond to each of the concerns in the review. For clarity of reference throughout this response, we denote the specific Weaknesses as W1, W2, etc. and Questions as Q1, Q2, etc.
>
> 1. **What constitutes the paper’s core innovation?** (W1, W2, Q2)
> We appreciate this opportunity to clarify our contributions. While our framework leverages existing tools, we respectfully posit that the core innovation lies not merely in the integration, but in defining a novel problem space and introducing specific reasoning mechanisms.
>     * **Novel Task Definition and Taxonomy:**
> Our primary contribution is the formalization of the "Interaction-Consistent Object Removal" task. Unlike general inpainting, this task requires resolving interaction dependencies between objects. We not only define this new problem but also systematically categorize it into four interaction types, establishing a clear taxonomy that makes this complex problem addressable and scalable for future research.
>     * **Methodological Innovations:**
> REORM is the first baseline specifically designed to tackle the new task. Beyond the pipeline structure, we introduce two specific algorithmic innovations:
> **Self-Correction Mechanism**: We introduce a verification mechanism where the MLLM generates a textual description of an "ideal post-edit state" to serve as a ground truth for evaluating the preliminary visual result. Leveraging the MLLM's extensive semantic priors is a novel approach to ensuring semantic fidelity.
> **MLLM-LLM Collaboration**: For our local variant, we design a unique collaboration between MLLMs and LLMs. We decouple visual perception (handled by MLLM) from complex logical reasoning (handled by LLM). By applying the complex text reasoning required for prompt chaining to the LLM, **we achieve superior interaction reasoning within limited resource constraints.
> Prior works such as BrushEdit [1] and Magicquill [2] don't possess these designs.**
>
> 2. **Runtime analysis and resource requirement per image for the overall pipeline** (W3, Q3)
> We appreciate the reviewer's query regarding the computational efficiency and resource requirements of our REORM framework. We have **included a runtime analysis in the updated PDF (Section 5.2).**
> As shown in **Table 1 ('Runtime' column)**, our approach exhibits a longer inference time compared to most other methods except for GPT Image 1. This is primarily attributed to the sophisticated and necessary reasoning process within our MLLMs, which is central to achieving high interaction consistency. The increased runtime stems from three key components:
>     * (1) The MLLM reason from the image to correctly identify all associated elements that should be jointly removed.
>     * (2) Ours(GPT-4o) incorporates an self-correction mechanism.
>     * (3) Multiple rounds of inference are performed by Ours(LLaVA + Llama).
>
>     While this sophisticated, reasoning-based process necessitates a longer runtime, this is a conscious trade-off for achieving substantial gains in the ability to handle complex removal sequences requiring deep semantic understanding. Our superior performance in interaction consistency justifies the increased computational overhead.
>     Furthermore, we explicitly acknowledge the concern regarding reproducibility, cost, and long-term accessibility of APIs. Addressing these issues was another motivation for developing our open-source, local deployment variant (utilizing LLaVA and Llama), as discussed in the Section 4.3.
>     **For the resource context**, both of our variants require a 24GB GPU. We conduct our experiment on an NVIDIA RTX 4090 GPU.
>
> 3. **Expansion of ICOREval evaluation dataset** (W4)
> We appreciate the reviewer's constructive suggestion regarding the dataset size and acknowledge that a larger benchmark strengthens the statistical robustness of our conclusions.
> Taking immediate action, we have **expanded ICOREval to 110 examples.** For context, **this scale compares favorably with or exceeds existing datasets in this domain**, such as RemovalBench (70), ObjectDrop (100), and OBER-Test (160).
> We have re-evaluated our method on this expanded dataset, and the updated quantitative results are presented in Table 1 of the revised PDF. **The results on the expanded set remain consistent with our initial findings, further validating the effectiveness of our approach.**

---

> > ### Comment · Reviewer_gsTw · 2025-11-26
> >
> > Thank you for the detailed rebuttal and clarifications.
> >
> > However, I still believe that the core contribution of this work remains incremental to ObjectClear. Prior works such as *ObjectClear* and *OmniEraser* have already claimed the ability to remove an object together with its associated visual effects, so this cannot be regarded as a novel task definition proposed by the authors. Although the paper provides a taxonomy, the overall pipeline still fundamentally relies on ObjectClear as a central component, augmented by an MLLM-based workflow. The proposed self-correction mechanism and MLLM–LLM collaboration appear to be engineering-oriented extensions rather than conceptual or methodological innovations.
> >
> > Therefore, I still maintain an original score of 4.

---

> > > ### Author Response · Authors · 2025-11-27
> > > **Clarifying Novel Contributions Relative to Prior Works**
> > >
> > > We thank the reviewer for the follow-up comments. We respectfully clarify that our contributions differ from prior works along two key dimensions: the scope of our task, and the role of the inpainting module in our framework.
> > >
> > > 1. **Visual Effects** vs. **High-Level Semantic Interactions**
> > > As discussed in Section 2 (Related Work: Object Removal, Line 129-135), while prior works like OmniEraser [Wei et al. 2025] and ObjectClear [Zhao et al. 2025] claim to remove objects with *"associated visual effects,"* their definition is strictly limited to low-level, passive elements (shadows and reflections).
> > >
> > >     Our defined task, Interaction-Consistent Object Removal, targets high-level interactions relationship. Crucially, our task explicitly encompasses complex categories defined in our taxonomy: **physically connected**, **target-produced**, and **contextually linked** interactions. These categories fundamentally do not belong to "visual effects". Handling these interactions requires deep **scene understanding** and **interaction analysis** to infer complex dependencies that may not be visible. This goes far beyond the capabilities of shadow/reflection removal models.
> > >
> > > 2. **Decoupling Our Framework from ObjectClear**
> > > The reviewer noted that our pipeline "fundamentally relies on ObjectClear." We would like to clarify that **ObjectClear is NOT a necessary or fixed component of our REORM pipeline**.
> > >
> > >     ObjectClear is used only as the current choice for the mask-guided removal step ("Mask-Guided Object Removal 1"). It is a passive tool that requires explicit object masks to function; therefore, it relies on our reasoning process to generate precise masks. Consequently, it can be seamlessly replaced by any other mask-guided inpainting model without altering our core reasoning methodology.
> > >
> > >     The **core contribution of REORM** lies in the MLLM-driven reasoning layer, which identifies the elements that must be jointly removed and orchestrates the entire multi-stage process. This layer enables **generalization** across interaction types and provides **interpretable**, step-by-step decision making. In contrast, the inpainting module functions as an interchangeable execution component rather than the conceptual foundation of the system. In this sense, **the reasoning layer serves as the “brain” guiding the “hand” (the inpainting tool)—a contribution fundamentally distinct from the tool itself.**
> > >
> > > We hope this clarification highlights that our work addresses a substantially broader semantic problem than prior shadow/reflection removal tasks and offers a generalized reasoning framework that remains independent of specific inpainting tools.

---

> ### Author Response · Authors · 2025-11-23
> **Our Reply to Reviewer gsTw (W5, Q1)**
>
> 4. **How is interaction consistency operationally defined or measured beyond qualitative examples? Could an explicit metric be introduced?** (W5, Q1)
> Thank you for the insightful comment. Because Interaction-Consistent Object Removal is a newly defined task, we initially followed evaluation practices commonly used in object removal research to ensure a fair and credible comparison. Consequently, we selected DINO, LPIPS, PSNR, and SSIM as our core metrics.
> Among them, the DINO Score serves as our primary quantitative proxy for assessing "interaction consistency." Since **DINO extracts deep semantic features, it effectively measures the semantic similarity between the edited result and the ground truth.** A high DINO score indicates that the model has correctly addressed the semantic context with the ground truth execution.
> We greatly appreciate the reviewer's constructive suggestion to introduce a more explicit metric. Inspired by your comment, we propose a "Label Consistency Score" based on object detection. This metric would utilize a pre-trained object detector to extract all object labels from both the edited image and the ground truth. We would then calculate the similarity to quantify whether the editing result contains the correct remaining objects and semantic layout.
> We agree that developing such metrics is a valuable direction, and we consider the “Label Consistency Score” a promising candidate for future standardized evaluation of the ICOR task.

---

### Comment · Area_Chair_Wrhz · 2025-11-25
**Discussion Period**

Dear Reviewers and Authors,

Thank you to the authors for submitting your rebuttal. We kindly encourage reviewers to take a moment to read the response and share any follow-up thoughts. Your timely engagement at this stage is highly valuable and helps ensure a fair, well-informed final decision.

We appreciate everyone’s efforts and contributions to the process.

Warm regards,
Your AC

---

> ### Comment · Reviewer_5SoM · 2025-11-25
>
> Dear AC and Reviewers,
>
> At this stage, my main remaining concerns are the scale of the ICOREval dataset and potential issues with diversity and bias. Although the authors have expanded the dataset to 110 examples, I am still unsure whether this scale is sufficient for a task with such complex interaction types.
>
> I would appreciate hearing your perspectives on this point, especially regarding whether the current dataset size and diversity are adequate to support the paper’s claims and serve as a meaningful benchmark for the community.
>
> Reviewer 5SoM

---

> > ### Author Response · Authors · 2025-11-27
> >
> > Thank you for the comment. Please refer to our response titled “Reply to Reviewer 5SoM: Diversity Analysis of the ICOREval Dataset” at the link below:
> > https://openreview.net/forum?id=q0cSTjtiLo&noteId=37SdKS6WRv

---

### Meta-Review · Area_Chair_wTmL · 2026-01-06

**Summary:**

The work is more like an integration of tools instead of a novel method/algorithm.

The proposed dataset -- ICOREval -- only contains 72 examples.

**Reviewer Concerns:**

The size of dataset is not addressed by authors.

**Reviewer Scores:**

N/A.

---

### Decision · Program_Chairs · 2026-01-26

Reject